# Robust Human Face Emotion Classification Using Triplet-Loss-Based Deep CNN Features and SVM

**DOI:** 10.3390/s23104770

**Published:** 2023-05-15

**Authors:** Irfan Haider, Hyung-Jeong Yang, Guee-Sang Lee, Soo-Hyung Kim

**Affiliations:** Department of Artificial Intelligence Convergence, Chonnam National University, Gwangju 500-757, Republic of Korea; irfan_haider99@hotmail.com (I.H.); hjyang@jnu.ac.kr (H.-J.Y.); gslee@jnu.ac.kr (G.-S.L.)

**Keywords:** emotion classification, SVM, triplet loss, transfer learning, ResNet18

## Abstract

Human facial emotion detection is one of the challenging tasks in computer vision. Owing to high inter-class variance, it is hard for machine learning models to predict facial emotions accurately. Moreover, a person with several facial emotions increases the diversity and complexity of classification problems. In this paper, we have proposed a novel and intelligent approach for the classification of human facial emotions. The proposed approach comprises customized ResNet18 by employing transfer learning with the integration of triplet loss function (TLF), followed by SVM classification model. Using deep features from a customized ResNet18 trained with triplet loss, the proposed pipeline consists of a face detector used to locate and refine the face bounding box and a classifier to identify the facial expression class of discovered faces. RetinaFace is used to extract the identified face areas from the source image, and a ResNet18 model is trained on cropped face images with triplet loss to retrieve those features. An SVM classifier is used to categorize the facial expression based on the acquired deep characteristics. In this paper, we have proposed a method that can achieve better performance than state-of-the-art (SoTA) methods on JAFFE and MMI datasets. The technique is based on the triplet loss function to generate deep input image features. The proposed method performed well on the JAFFE and MMI datasets with an accuracy of 98.44% and 99.02%, respectively, on seven emotions; meanwhile, the performance of the method needs to be fine-tuned for the FER2013 and AFFECTNET datasets.

## 1. Introduction

Humans can communicate in a variety of ways, including through words, gestures and feelings. A comprehensive and accurate comprehension requires one’s own sentiments and the hidden meanings it carried. Integration of these characteristics into machines that allow for a diverse and natural style of communication has become an attractive area of research in robotics, particularly in the realm of humanoid robots. Emotions are of several types, i.e., happy, sad, wonder, fear, guilt, confused and shocked. Mostly, six kinds of emotions are used. Emotions are detected for different purposes, i.e., healthcare apps, chat-bot apps and to make the system intelligent. EmotiChat [1] is a tool that is designed in this fashion. It is a chat program that can read your feelings. When the computer detects a positive emotion (a grin, for example), an emoticon is added to the conversation window. It may be useful in areas apart from human–computer interactions (HCI), such as surveillance or driver safety. Automatic systems can be made more responsive if they can gauge the driver’s emotional state.

Recognizing a person’s emotions based on their facial expressions is an integral part of interpersonal communication since they are an intuitive representation of a person’s mental state and carry a wealth of emotional information. Zeng Guofan [2] was a well-known ancient Chinese thinker who developed techniques for emotion recognition from people’s faces. Today, researchers use various methods to detect facial expressions and to recognize emotions. Features can be extracted manually [3] or by an automatic detection method [1]. Action units (AU) are used to define facial expressions in the Facial Action Coding System (FACS). Raising one’s inner brow is an example of an action unit. The emotions on someone’s face are described by the simultaneous activation of many AUs [4]. To be able to make a claim regarding the degree of activation of the related emotions, it is helpful to be able to accurately recognize AUs. Facial landmarks can be hand-crafted, as in the work of Kotsia et al. [3]. It can be challenging to spot such features since their relative distances vary from person to person [5]. Besides emotions, AUs can also be used to identify facial textures. Filters can be used to identify the structural changes brought on by the expression of emotion on a person’s face [5].

Face extraction from images is a crucial component of several approaches. The use of convolutional neural networks (CNNs) was validated for emotion recognition and feature extraction. Softmax [6] is often used to quantify the difference between the CNN’s outputs and the supervised signal, and the triplet loss [7] and its variations were suggested to further impose intra-class compaction and inter-class distinctiveness of the learned features. We propose the quadruplet loss [8] as an enhancement of the triplet loss to further separate the classes. Adjusting the triplet loss’s selection and margin specifications is a major focus for improving its functionality. During network optimization, the most difficult sample triplet is chosen to reduce intra-class distance and to increase inter-class distance. Nonetheless, the network is better able to learn from the most challenging data triplets due to these triplet losses. Using either intra-class distance or inter-class distance, the most difficult couples are selected as training samples by Song et al. [9]. convolutional neural network (CNN) models are very helpful in detecting emotions and feature extraction. CNN can achieve good performance in image recognition and attains higher recognition accuracy than conventional algorithms. CNN models also showed good results in detecting emotions by using EEG signals. In our study, we present primarily related material to the suggested method, as well as a few carefully chosen publications that provide an overall perspective on the various approaches.

However, current CNN models are not good enough. In our study, we customized ResNet18 to enhance the accuracy of emotion recognition. Our customized model performed well on two datasets, i.e., JAFFE and MMI. The main contribution of our model is to detect the emotions accurately and to customize ResNet18 to achieve better results than others.

## 2. Literature Review

Caleanu [10] and Bettadapura [11] provided an in-depth analysis of the field of facial expression recognition. Many detectors were used for recognizing emotion and for detecting faces from images or videos. Sarah et al. [12] used YOLO in their study for detecting faces from images. YOLO has many versions and is mostly used for the object detection task, and a face is also an object, which is it was used for emotion detection tasks. Liu et al. [13] used retina faces to detect faces from images, which he also used for aligning, analyzing and cropping the images into a size of 256 × 256. In one study [14], faces were detected from video by using retina face detectors. Happy and Routray [5] distinguished between six fundamental emotions using the Extended Cohn–Kanade Dataset. Their method detected the facial area in an image and returned its coordinates. Facial features, such as the eyes or lips, are identified and designated from a specific area. Features are extracted from regions that show the largest contrast between two images. Firstly, reducing the features’ dimensionality, we may then feed them into a support vector machine (SVM). A 10-fold cross-validation is used to test the efficacy of the procedure. The mean precision is 94.09%.

Byeon and Kwak [15] proposed using a video to detect emotional states. A 3D convolutional neural network (CNN) that takes input from sequences of five frames was constructed. An accuracy of 95% was achieved using a database of only 10 people. Song et al. [16] employed a deep convolutional neural network for face emotion recognition. A neural network was built with a total of 65 thousand neurons for cascading five layers. An accuracy of 99.2 percent on the CKP set was accomplished with the help of convolutional, pooling, local filter, and one fully connected layer. Overfitting was prevented by employing the dropout technique.

The Extended Cohn–Kanade dataset was developed by Luecy et al. [17]. Annotations for both emotions and action units are included in this dataset. They also tested the datasets for classification accuracy using active appearance models (AAMs) in tandem with support vector machines (SVMs). They used AAM, which creates a mesh from the face, to pinpoint its location and to follow it across photos. They used this mesh to generate two feature vectors. To begin, the axes of rotation, translation, and scaling were applied to the vertices. Second, a grayscale image was derived from the mesh information and the input pictures. With the help of a cross-validation approach in which one participant is removed from the training phase, they were able to increase the accuracy to over 80%.

An emotion recognition system based on facial expressions was created by Anderson et al. [1]. There are three main parts to their system. The first is a tool developed for locating faces, called a face tracker (a ratio template derivative). The second part is a facial motion-tracking optical flow algorithm. The final piece is the recognition engine itself. It uses SVMs and multilayer perceptron (MLPs) as its foundational neural network models. EmotiChat takes this strategy and runs with it. Their rate of success in recognizing patterns is 81.82%.

Emotion recognition by using features extraction has some limitations due to complex facial structures. To address this issue, a unique feature fusion dual-channel expression recognition algorithm was proposed in this paper using principles from machine learning and philosophy.

In particular, the issue of subtle variations in facial expression is not taken into consideration by the feature generated by a convolutional neural network (CNN). The proposed algorithm’s initial fork takes as input a Gabor feature extracted from the ROI region. To make the most of the operational facial expression emotion region, it is important to first segment the area from of the main facial images before applying the Gabor transform to extract the area’s emotion properties. Specific attention must be given to describing the surrounding area. The alternative method recommends a channel attention network that uses depth separable convolution to improve linear bottleneck structure, reduce network complexity, and safeguard against overfitting by way of the creation of a well-designed attention module that considers both depth of the convolution layer and spatial information. For the FER2013 dataset, it surpasses the competition by a wide margin due to its increased focus on feature extraction, which in turn leads to better emotion recognition accuracy.

Kotsia and Pitas [3] developed a method for emotion detection in which a candidate grid, a face mask with a small number of polygons, is superimposed on a person’s face. The grid is superimposed on the image in a completely random location and then must be moved to the subject’s face by hand. As the feeling progresses, a Kanade–Lucas–Tomasi tracker keeps tabs on the grid’s position. For multiclass SVMs, the grid’s geometric displacement information serves as the feature vector. Those feelings include anger, repugnance, fear, joy, sorrow, and surprise. They ran tests on the Cohn–Kanade dataset and found that their model has a 99.7 percent success rate. Based on local binary patterns (LBPs), Shan et al. [18] developed a system for emotion recognition. Facial area LBP features are extracted from the input face image. AdaBoost is used to locate the image regions that are most informative for classification. After testing numerous classification methods, we found that SVMs using Boosted-LBP features had the highest recognition accuracy (95.1% on the CKP dataset).

Using robust normalized cross-correlation (NCC), Zafar et al. [19] suggested an emotion recognition system in 2013. The “Correlation as a Rescaled Variance of the Difference between Standardized Scores” normalized coefficient calculator (NCC) was utilized here. Pixels that strongly or weakly influence the template matching are deemed outliers and are ignored. This method was evaluated using both the AR FaceDB (85% recognition accuracy) as well as the Extended Cohn–Kanade databases (100% recognition accuracy).

### 2.1. Face Detection

The numerous real-world applications of face recognition have attracted a lot of research and development attention. In the days before the deep convolutional neural network (deep CNN), face detection depended primarily on characteristics that were created by hand [20]. Researchers presented a plethora of strong, manually created features, including HAAR, histogram of oriented gradients (HOG), LBP, scale-invariant feature transform (SIFT), and aggregate channel features (ACF). However, deep CNN has greatly outpaced the performance of these feature extractors. The last several years have seen an expansion of models, with deep CNN proving particularly effective at a wide variety of target identification applications. We characterize the target identification tasks such as a pair of tasks, one involving the classification of target candidate regions and the other using their regression. Object detection networks come in a wide variety; some examples include the RCNN family [20], SSD [21], YOLO series [22], FPN [23], MMDetection [24], EfficientDet, transformer (DETR), and Centernet.

Even though there are unique difficulties in detecting faces, such as multiscale, small face, low light, dense scene, etc., inter-class [25,26,27] is also an issue in image classification that misclassifies the emotions because of having the same person’s face in different classes, as can be seen in the example of having the same face in Figure 1 and Figure 2. These problems are the same to those faced when searching for any object as shown in Figure 3. For this reason, broad object detection networks can be used as a stepping stone for training face detection networks. Scale, position, occlusion, expression, and cosmetics are all areas that might cause issues on their own. Researchers have come up with a number of approaches to these problems, including the Cascade CNN, multi-task cascaded convolutional networks (MTCNN), and the single-stage headless face detector (SSH).

#### 2.1.1. YOLO

The object detection community has contributed significantly to the evolution of the YOLO algorithm over the past five years, resulting in version 5. The creator of the YOLO algorithm proposed the first three versions, which are referred to as YOLOv1 [28], YOLOv2 [29], and YOLOv3 [24]. YOLOv3 is widely regarded as a major advancement in terms of performance and speed. The Softmax loss is replaced with the binary cross-entropy loss, and it identifies multiscale features (FPN) [23] and a superior backbone network (Darknet53). A separate research group unveiled YOLOv4 [30] in 2020’s first few months. The group gave serious thought to a wide variety of YOLOv3 [24] variants, including the backbone along with the alleged bags of freebies and bags of specialties. A month later, a new research group unveiled YOLOv5 [31], which was substantially smaller in size, faster in speed [30], and fully implemented in Python (PyTorch). To this point, the field of object detection has greeted it with open arms.

#### 2.1.2. Haar Cascade

To detect objects, the Haar classifier relies heavily on these Haar-like properties. These characteristics do not rely on pixel intensity but rather on the contrast shift between neighboring squares. Variations in brightness and darkness are identified based on the contrast differences between clusters of pixels. A Haar-like feature is formed by two or three neighboring groups that share a similar contrast variance. They use Haar-like features to detect images [32]. Raising or lowering the size of the pixel group that is being analyzed is a simple way to adjust the scale of Haar characteristics. This paves the way for characteristics to be utilized for detecting objects of varying sizes.

Haar classifier cascades must be trained before they can accurately detect human facial characteristics such as the mouth, eyes, and nose. Training classifiers requires the implementation of the mild AdaBoost algorithm and the Haar feature methods. Intel created Open Computer Vision Library, an open source library specifically designed to simplify the incorporation of programs linked to computer vision (OpenCV). Detection and training of Haar classifiers, such as implementation applications in HCI, robotics, authentication, computer vision, and other domains where visualization is essential, can make use of the OpenCV library. Ref. [32] is only one example. We need two separate photo collections to train the classifiers. Images and videos that do not feature the sought-after item form one set (a face, in this example). The term “negative images” is used to describe this group of pictures. The other group of pictures, the positive ones, show the object in use. Image name, top left pixel, height, and breadth of the object specify where the objects are located within the positive photos. Five thousand low-resolution photos (at least a megapixel) were utilized to train the face characteristics. Among these were pictures of paperclips and photographs of forests and mountains.

#### 2.1.3. Retina Face

The cutting-edge RetinaFace facial detector for Python is based on deep learning and includes facial landmarks. When used in a crowded environment, its detection performance stands out as exceptional. To accomplish pixel-wise face localization, RetinaFace is a one-shot framework that employs three subtasks. Face detection, 3D reconstruction using a mesh decoder, and 2D alignment are the tasks at hand.

### 2.2. Loss Functions

#### 2.2.1. Triplet Loss Function

For the objective of understanding the emotions expressed in a human voice, a neural embedded system that depends on triplet loss and residual learning was proposed by Kumar et al. [7]. The proposed approach uses the emotional tone of the speaker’s words to learn the optimal embedding.The triplet loss function takes in an anchor image, a positive image, and a negative image as inputs. An already-labeled image serves as the “anchor”, while an image with the same label as the anchor and an image with a different label serve as the “positive” and “negative”, respectively. Discovering an embed of features for an image where the distance between the anchoring and the positive image is smaller than the distance seen between the anchor and the negative image is the goal of the triplet loss function. The triplet loss function can be written as:$$L=max(0,|\;|\;f\left(x_{a}\right)-f\left(x_{p}\right)\left|\;\right|{}^{2}-|\;|\;f\left(x_{a}\right)-f\left(x_{n}\right){\left|\;\right|}^{2}+margin)$$
where f is the feature embedding function, $x_{a}$ is the anchor image, $x_{p}$ is the positive image, and $x_{n}$ is the negative image, | |.| | is the $L_{2}$ norm, and margin is a hyper-parameter that controls the margin between the positive and negative distances. Zhang et al. [8] provided identity loss in a supporting task to monitor the training process and to improve the durability of the elementary task without interrupting the entire training stage for facial expression recognition (FER) with deep metrics.

In order to bring images that have the same label closer to one another in feature space, the neural network is programmed to minimize the triplet loss function. After the network was trained, features embedding may be used to classify facial expressions by passing the image through it and by letting a straightforward classifier, such as a support vector machine classifier, make a prediction about the expression label. When using a triplet loss function, the distance among objects of various classes in the feature space is effectively increased [33]. To further improve the effectiveness of our method, we incorporated the triplet loss function.

#### 2.2.2. Cross Entropy

Cross-entropy loss, also called log loss, is used to measure the efficiency of a classification process whose output is a probability value between 0 and 1. Cross-entropy loss increases as the difference between the expected probability and the actual label grows larger. The cross-entropy loss function and the loss function are both proposed in this method [34]. We will then evaluate the algorithm’s performance based on the precision and recall of its detections. Simulation findings show that the proposed technique outperforms state-of-the-art classifiers such as CNN in terms of prediction accuracy.

### 2.3. Facial Emotion Classifiers

#### 2.3.1. Linear Discriminant Analysis (LDA)

Linear discriminant analysis (LDA) employs a linear regression model for categorization and dimensionality reduction [35]. Features extracted from pattern categorization issues are its primary use. After its initial development by Fisher in 1936 for two classes, the linear discriminant was later generalized by C.R. Rao for many classes in 1948. By utilizing LDA to project data from a D-dimensional feature space onto a D′ (D > D′) dimensional space, we may maximize variation across classes while decreasing variability within classes. Logistic regression, one of the most popular linear classification models, does well for binary classification but poorly for multiple classification problems with clearly distinguishable classes. LDA can be used in data preprocessing in the same way that principal component analysis (PCA) can be used to minimize the number of characteristics and hence the computing expenses. Face recognition algorithms also make use of LDA. Similarly, LDA is used in face detection methods.

#### 2.3.2. Support Vector Machine (SVM)

Invention of SVM can be credited to Vapnik et al. [36], with subsequent modifications by Corinna Cortes and Vapnik [10]. SVM is employed in data analysis and pattern recognition. Yag et al. mention [37] utilized support vector machines to categorize plant diseases. Pattern regression analysis and classifications are common applications for SVM, which is a supervised learning algorithm that infers a function or relationships from the given training data. In order to efficiently assign a new example point to one of two classes, an SVM algorithm requires a training set of examples from each class.

#### 2.3.3. Softmax Regression

Assumptions for every class label are provided by Softmax classifiers, while the margin is provided by the hinge loss metric. As human beings, we can more easily grasp probabilities than range scores (such as in hinge loss and squared hinge loss). Neural network models that forecast a multinomial probability density function use the Softmax function as the input signal in the output layer. In other words, when determining class membership on more than two class labels, Softmax is utilized as the activation function.

## 3. Proposed Methodology

In this section, we first describe the overall proposed pipeline with a face detector to detect and tighten the face bounding box and a classifier to obtain the facial expression class of detected faces using deep features from a ResNet18 trained by using triplet loss. We use RetinaFace to detect and cut out the detected face regions from the original image for each input. Then, we train a ResNet18 model using cropped face images with triplet loss and use this model as a feature extractor. Afterward, deep features generated from the feature extractor are fed into an SVM classifier to produce the facial expression class of the face. A frame extractor was developed to process video input to extract the twenty representative images. The proposed pipeline is defined in Figure 4.

### 3.1. Frame Extractor

The proposed pipeline’s encoder is designed for image input only. To eliminate this weakness and to make the pipeline work for image input and video, we constructed a frame extractor to extract the most representative frames of each video by considering the problem as anomaly detection on a time series.

Assume that we have a n−frame video input with resolution of W×H×C, which can be represented as $X={\left\{X_{t}\right\}}^{n}$ for $X_{i}\in R^{W\times H\times C}$. We aimed to construct a 1−D time series $T={\left\{T_{t}\right\}}^{n}$ for $T_{t}\in R$ from this input and tp detect the outliers in the series. First, we applied a Gaussian filter $G\left(x,y\right)=\frac{1}{2\pi \sigma^{2}}\exp\left(-\frac{x^{2}+y^{2}}{2\sigma^{2}}\right)$ on each frame $X_{t}^{{}^{\prime }}=G*X_{t}$ with ∗ as the convolution operator to remove the effect of noise between frames. Then, we applied a simple sum operator over $X_{t}$ dimensions to map between $T_{t}$ and $X_{t}$ by $T_{t}=\sum X_{t}$. After obtaining the T series, we removed the first and last 20 values because they regularly represent a neutral facial expression. Then, the differencing method was applied to compute the difference between frames with a set of lag configs. With m−lag config, we obtained the new $T^{{}^{\prime }}$ series with $T_{t}^{{}^{\prime }}=T_{t}-T_{t-m}$. In the last step, the outliers were detected using the mean absolute deviation method [38]. Across lag configs, we removed the duplicated frames and obtained the twenty unique and most representative frames. An example is shown in Figure 5.

### 3.2. Face Detector

We only need the human face image regions to know which facial emotion is expressed. Images with complex backgrounds and small faces can affect the performance of the facial expression recognition pipeline. We decided to plug in a face detector to detect only face regions in each image and to remove unrelated regions that can harm our pipeline performance.

We surveyed many existing face detectors, such as RetinaFace [39] and YoloFace [40], to obtain the best face detector for our pipeline. All face detectors were compared using the WiderFace dataset [41]. We finally chose RetinaFace because it has the best performance based on the WiderFace dataset.

With the RetinaFace model, we cropped face regions on each candidate image, obtained only the largest detected face, and kept it for the latter pipeline component input, as shown in Figure 4. The face detector pipeline can be seen in Figure 6.

### 3.3. Deep Feature Extractor

After receiving a proper dataset with cropped faces, we used a convolutional neural network [42] to extract the deep features of each image. A CNN model can contain many types of layers, such as convolution, activation, and pooling layers. Each layer extracts higher features than the previous layer [39]. In order to obtain the best result, we applied the advanced CNN architect proposed in [43]. With ResNet [43], we can stack more convolution layers to the model without worrying about the vanishing gradient problem, and the model can achieve higher performance.

After the last pooling layer of the CNN model, we added a fully connected layer that outputs a 128−D embedding vector or deep features for each input. Then, this embedding vector is normalized to a unit sphere as in [44] before being fed into any loss functions or classifiers. Following the FaceNet paper [44], we used the triplet loss function to optimize the extracted embeddings directly. We adopted the online triplet mining technique introduced in [45]. This technique enables us to generate triplets based on data batches in each learning iteration. The whole training diagram is shown in Figure 7.

### 3.4. Facial Expression Classification

We also used many different classifiers (SVM, LDA, Softmax Regression) to obtain the facial expression class. At the end of each training iteration, the produced embeddings with labels of their original images were used to fit classifiers. Once trained, these classifiers were evaluated on the validation and test set. In this work, we used scikit-learn’s implementation of three classifiers.

## 4. Results and Evaluation

### 4.1. Datasets

**JAFFE** [46]: This database was introduced by Michael Lyons, Miyuki Kamachi, and Jiro Gyoba [46] and has a total of 213 images. In this datasets=, the number of subjects is 10, and the images are grayscale in the size of 256 × 256. The photos were annotated, with average semantic scores for each facial emotion, by 60 annotators after each subject performed seven facial expressions (six basic and one neutral).

**FER2013** [47]: This dataset contains images and categories describing the emotion of the person in it. The total number of grayscale images in the size of 48 × 48 in the dataset is 35,953 images in seven different facial expression classes. The dataset has three subsets for training, validation, and testing. While analyzing the dataset, we found that the dataset has a significant data imbalance problem due to the “disgust” class only having a few samples compared to the others. FER2013 distribution is depicted in Figure 8.

**AFFECTNET** [48]: This dataset includes samples of various sizes and excellent images in RGB or grayscale-based color. There are eight distinct classes (surprise, angry, sad, contempt, disgust, fear, neutral, and happy). The validation and training sets are divided as of the FER2013. Similar to the FER2013 dataset, AFFECNET also suffers from the data imbalance problem, but this is due to the dominance of the number of the “happy” class. AFFECTNET distribution is depicted in Figure 9.

**MMI** [49]: This is a database containing 236 videos of people showing emotions with annotations. Twenty frames from each video that best capture its content were automatically extracted by the frame extractor discussed in Section 3. A total of 4756 photos of frontal views of the human face was retrieved.

### 4.2. Experiment Design

In order to assess the effectiveness of the proposed pipeline, we evaluated its performance on the JAFFE, FER2013, AFFECNET, and MMI datasets. For better classification, we obtained optimized deep features at the "avg_pool" layer of the ResNet18 model and fed it into SVM for facial emotion prediction. The proposed approach was validated on the four standard datasets, JAFFE, FER2013, Affectnet and MMI. The developed TLF-ResNet18 model outperformed on the JAFFE and MMI datasets and offered comparable results on the FER2013 dataset. The proposed model outperformed the state-of-the-art approaches, which show its effectiveness. The developed pipeline is flexible and can be extended to address more facial expressions. In all cases, we measured the benefit of the proposed method compared to other previous results. As seen below, our pipeline outperforms many current SoTA results.

All experiments were set up using ResNet18 as the feature extractor to generate the unified embeddings from original human face images. We applied basic data augmentations to improve generalization performance, such as horizontal flip, vertical flip, random rotation, and random crops. All images were normalized to grayscale-based images. We found that 48 × 48 is the optimal image size when considering both speed and accuracy for facial expression recognition; increasing the image size does not increase the accuracy much. We also implemented a weighted data sampling method that will assign a smaller weight to all samples in a class with a larger total number of samples. In each learning iteration, the data loader prioritized creating batches of data on samples that have a larger weight. This method greatly improved the imbalance datasets such as FER2013 and Affectnet.

We used Adam [50] as the optimizer to optimize the loss value with a learning rate of 0.001, which is reduced by a factor of 0.9 if there is no improvement in validation accuracy in the last five epochs. We ran the training for 100 epochs and evaluated the learned embeddings at the end of each learning iteration by fitting the SVM, LDA, and Softmax regression classifiers. All experimental results can be seen in Figure 10. We present and discuss in detail each case in the next subsections.

### 4.3. Small and Balanced Datasets: JAFFE, and MMI

JAFFE and MMI are small datasets for facial expression recognition and do not have a separate evaluation dataset. We split these datasets randomly with a ratio 7/3 of training and validation sets to evaluate the unified embeddings on these datasets. Specifically, the JAFFE dataset that has a total of 214 samples in seven emotion classes is split to have 149 samples in the training set and 65 samples in the validation set; finally, the MMI dataset is split to have 3329 samples in the training set and 1427 in the validation set.

The training curve of these three datasets can be seen in Figure 11 and Figure 12. The results of classifiers have no difference and demonstrate SoTA performances. They also show that our method is robust and has achieved the ultimate performance that could not be better. The best accuracy of the proposed method and previous works on the JAFFE and MMI dataset are shown in Table 1 and Table 2.

### 4.4. Imbalance Dataset: FER2013 and AFFECTNET

In contrast, the FER2013 and AFFECTNET datasets are much larger than previous ones, with a total of 26,899 and 1,749,905 samples, respectively. In addition, these two datasets have a problem of imbalanced data samples between classes. Figure 8 and Figure 9 show the sample distribution of the FER2013 and AFFECTNET datasets that reveal a serious imbalance problem in both datasets.

We trained the feature extractor ResNet18 with 300 epochs on FER2013 and 12 epochs on AFFECTNET; other training configurations are kept the same as the experiments on the JAFFE and MMI datasets. The training curve is depicted in Figure 13 and Figure 14. All classifiers are not able to obtain good results on these datasets, with approximately 74.64% and 62.78% in accuracy, and they are comparable with the current SoTA results, with 75% on FER2013 and 3.03% on AFFECTNET. For better analysis, we provide confusion matrices and ROC curves of the best model on each dataset in Figure 15 and Figure 16, respectively.

In FER2013, the “happy” and “sad” emotion classes have the best performance with 91% and 83% in accuracy, respectively. They also have the largest AUC compared to others in ROC figures. In contrast, the “fear” class has the worst performance, and samples in these classes are usually misclassified to each other. The “disgust” class has very few samples in the FER2013 dataset but achieves an acceptable performance with 73% accuracy.

In AFFECTNET, the “happy” emotion class has a dominant number of samples in it and has achieved the best performance in the AFFECTNET dataset with 77% in accuracy. The worst performance is in the “neutral” class, with only 54% in accuracy, and its samples are usually misclassified with the “contempt” class.

We also provide the best accuracy of the proposed method and comparison of previous works in Table 3 and Table 4.

## 5. Ablation Study

In this section, we investigate the effect of different training methods on model performance. Table 5 and Table 6 show that the triplet loss and weighted data sampling methods have significant impact on the model performance when training on imbalanced datasets such as FER2013 and AFFECTNET.

### 5.1. Using Cross-Entropy Loss

We conducted some experiments by using other loss functions such as cross-entropy loss instead of triplet loss to justify the effectiveness of our proposed method.

Figure 17 and Figure 18 show that cross-entropy loss generates worse results on the FER2013 and AFFECTNET datasets than the proposed method, with only 74.1% and 60.70%, respectively. This indicates that the proposed method can produce deep features with stronger discriminative power.

### 5.2. Effect of Weighted Data Sampling Method

As described in Section 4.2, we implemented and applied the weighted data sampling method to deal with the data imbalance problem. We found that the data imbalance problem in FER2013 and AFFECTNET has a strong negative effect on model performance. In this subsection, we provide experimental results without applying weighted data sampling in Figure 19 and Figure 20.

## 6. Discussion

Our proposed method uses triplet loss to enable the model to generate deep features for each image input. With this loss function, we may bring together, in latent space, the deep features of every pair of samples from the same class and perform the opposite for every pair of samples from different classes. As a result, our model is better able to discriminate across classes, leading to improved accuracy even on datasets with high inter-class similarity. The triplet loss makes the deep features of each sample pair of a same class closer in latent space and vice versa for each data pair of two different classes. The method also achieves SoTA performance without the need for face registration, extensive data augmentation, or extra training data or features. Our FER method is conceptually easier to understand than earlier methods because it does not require face registration and is not impacted by registration errors. We anticipate that using supplemental training data and thorough FER-specific data augmentation would considerably enhance the performance.

Our approach already outperforms earlier efforts on FER and achieves the best results on the JAFFE and MMI datasets. Our approach is most effective for those datasets that have inter-class similarity. To obtain better results, however, the performances on FER2013 and AFFECTNET need to be severely fine-tuned, as they are still far from SoTA performance. This paper proposes a robust pipeline and studies the FER performance on many FER datasets with different characteristics (from small and balanced datasets, such as JAFFE and MMI, to larger and imbalanced datasets such as FER2013 and AFFECTNET). However, due to dataset bias, the results from this and other FER datasets are only representative of real-world FER performance. This restriction applies to FER research in general as well as to this particular study.

We further speculate that the decreased performance compared to state-of-the-art approaches is due to the absence of fine-tuning on the FER2013 and AFFECTNET datasets. Additional training data and comprehensive, FER-specific data augmentation, as proposed in the research, are expected to greatly improve performance on these datasets.

## 7. Conclusions

In this paper, we proposed a method to solve inter-class problems in emotion recognition by using images and videos. Our pipeline uses RetinaFace as a face detector to detect and crop human face regions in images in order to remove unnecessary information from the input. We also implemented a frame extractor that is based on the MAD outlier detection method in time series to extract the most representative frames in videos. Then, each human face region was fed into a feature extractor ResNet18 model to produce deep features, and the deep features were fed into an SVM classifier to obtain the facial expression class. Performance on other classifiers, such as LDA and Softmax regression, are also reported in this paper. We achieved state-of-the-art curacies on two datasets named JAFFE and MMI, with comparable results on the FER2013 and AFFECTNET datasets. Multimodal approaches that combine image- and video-based methodologies with other modalities such as voice, text, or physiological inputs should be examined to improve the accuracy of emotion recognition in future research. These methods may improve the overall performance of emotion detection systems by leveraging multiple sources of information, making them more resilient to different environmental conditions.

## Figures and Tables

**Figure 1 sensors-23-04770-f001:**
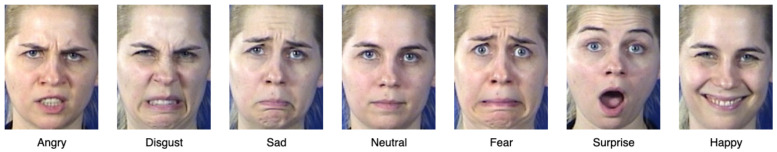
Inter-class sample.

**Figure 2 sensors-23-04770-f002:**
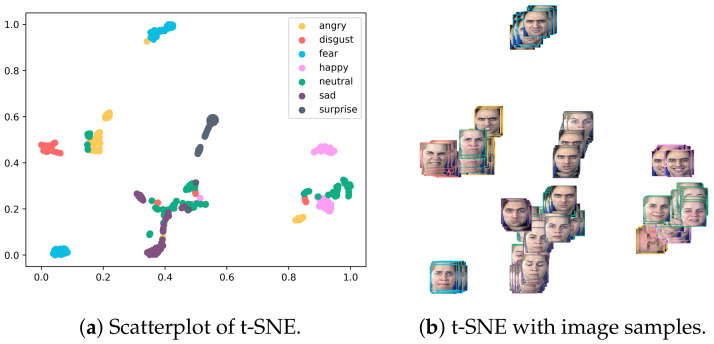
t-SNE visualization.

**Figure 3 sensors-23-04770-f003:**
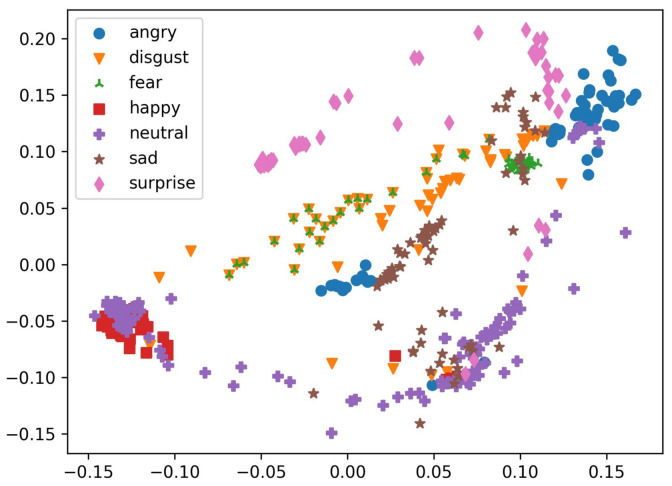
Scatterplot.

**Figure 4 sensors-23-04770-f004:**
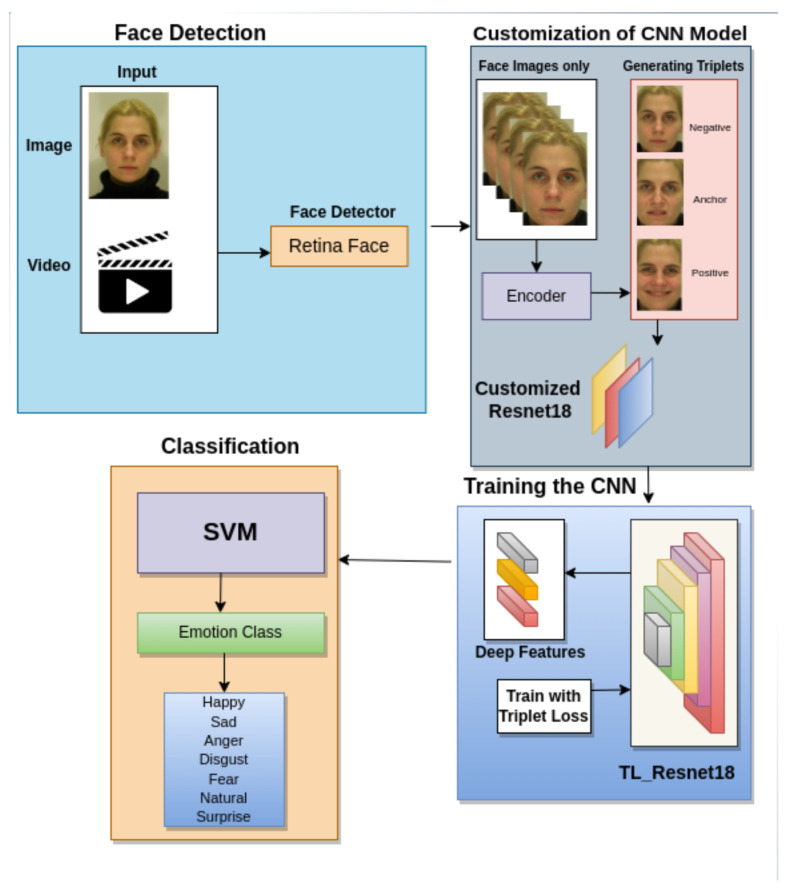
The proposed TLF-ResNet18 pipeline for facial expression recognition with a face detector to detect and tighten the face bounding box and an SVM classifier to obtain the facial expression class of detected faces using deep features from trained ResNet18.

**Figure 5 sensors-23-04770-f005:**
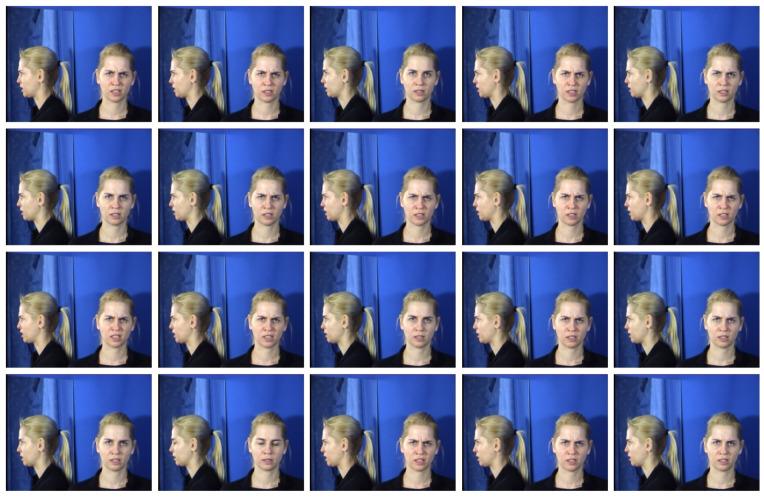
The twenty unique and most representative frames extracted from an angry expression video of MMI database.

**Figure 6 sensors-23-04770-f006:**
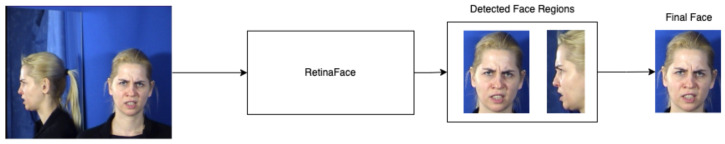
The face detector pipeline with multiple faces in the input image and output as the largest face only.

**Figure 7 sensors-23-04770-f007:**
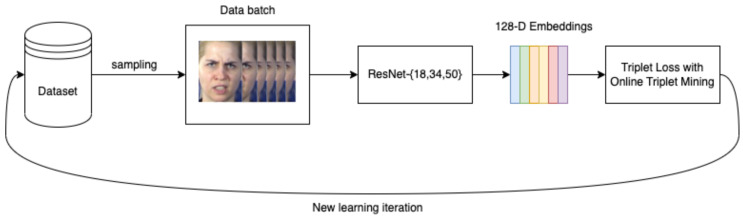
Training diagram with online triplet mining technique from [9].

**Figure 8 sensors-23-04770-f008:**
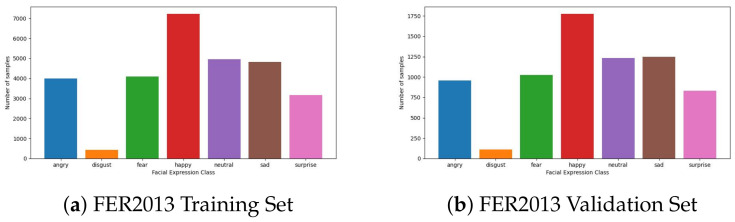
Dataset distribution of the training and validation set of FER2013.

**Figure 9 sensors-23-04770-f009:**
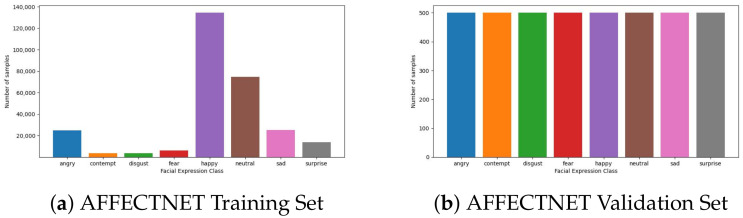
Dataset distribution of training and validation set of AFFECTNET.

**Figure 10 sensors-23-04770-f010:**
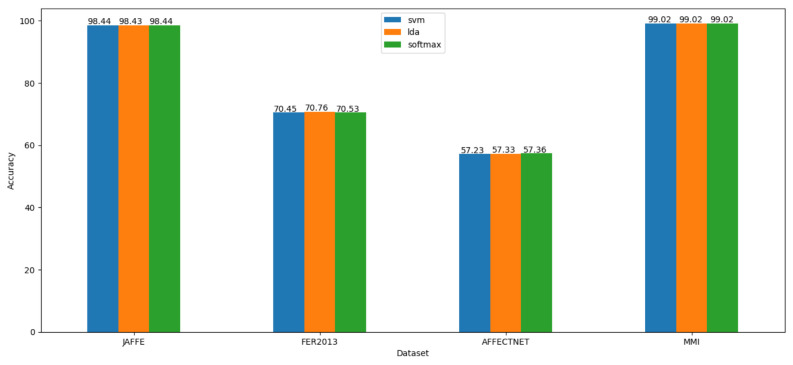
The best accuracy of each classifier when trained and evaluated on embeddings learned on each dataset.

**Figure 11 sensors-23-04770-f011:**
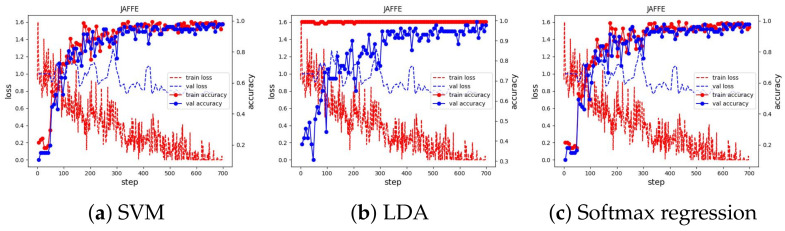
Training curve of the three classifiers on the JAFFE dataset.

**Figure 12 sensors-23-04770-f012:**
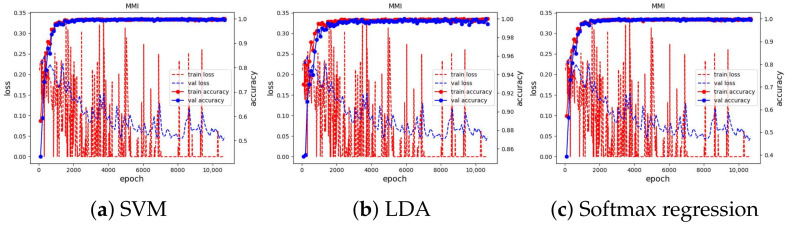
Training curve of three classifiers on MMI dataset.

**Figure 13 sensors-23-04770-f013:**
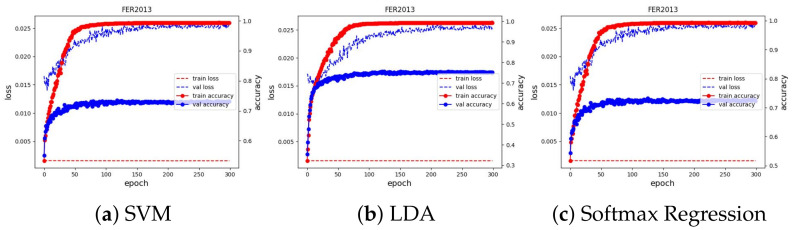
Training curve of three classifiers on the FER2013 dataset.

**Figure 14 sensors-23-04770-f014:**
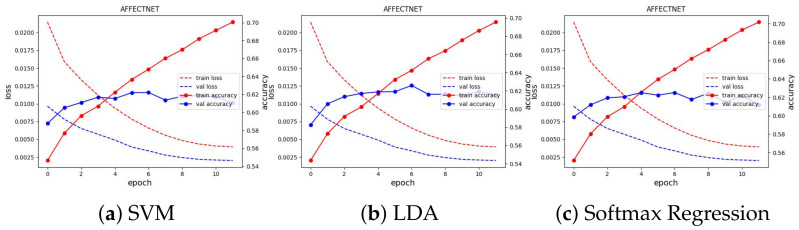
Training curve of three classifiers on the AFFECTNET dataset.

**Figure 15 sensors-23-04770-f015:**
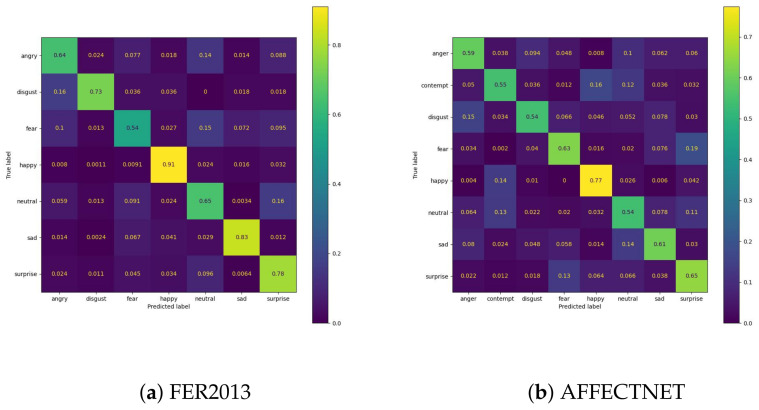
Confusion matrix of best models on the validation set of each dataset.

**Figure 16 sensors-23-04770-f016:**
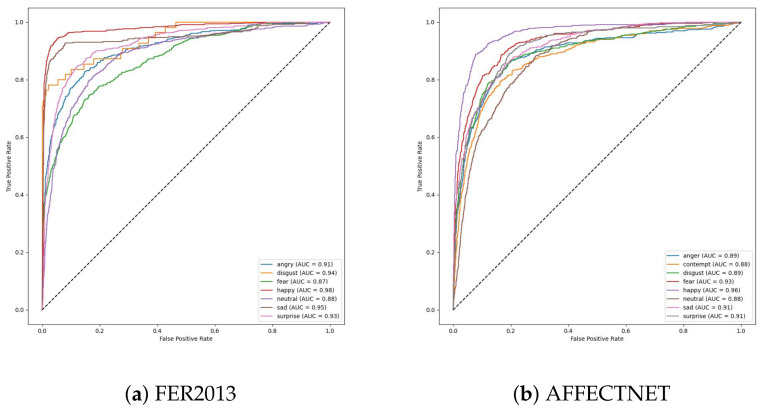
ROC curve of best models on the validation set of each dataset.

**Figure 17 sensors-23-04770-f017:**
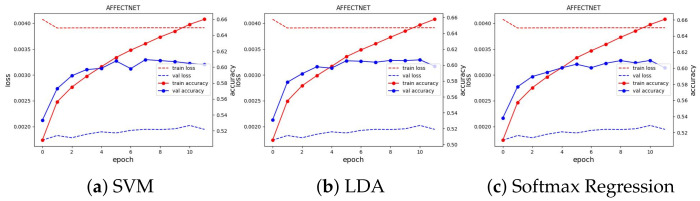
Training curve of three classifiers using Cross Entropy loss on the AFFECTNET dataset.

**Figure 18 sensors-23-04770-f018:**
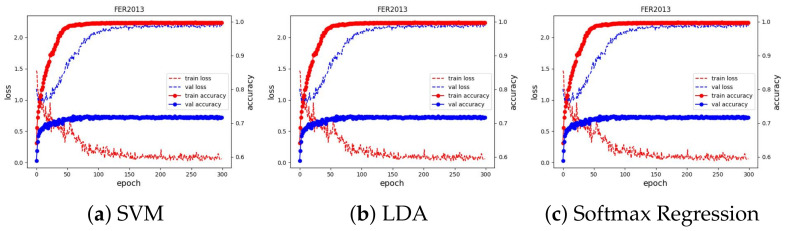
Training curve of three classifiers using Cross Entropy loss on the FER2013 dataset.

**Figure 19 sensors-23-04770-f019:**
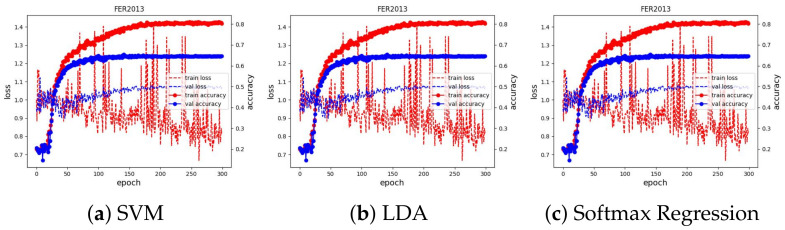
Training curve of three classifiers on the FER2013 dataset without applying a weighted data sampling method.

**Figure 20 sensors-23-04770-f020:**
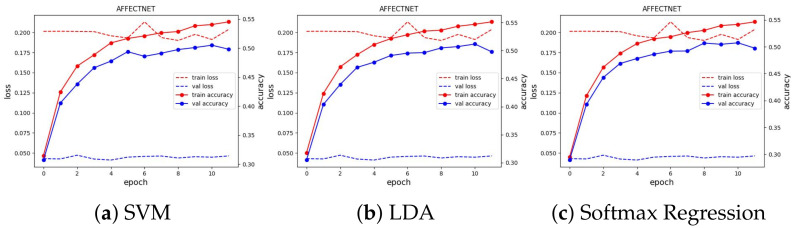
Training curve of three classifiers on the AFFECTNET dataset without applying a weighted data sampling method.

**Table 1 sensors-23-04770-t001:** Performance comparison between our proposed pipeline with previous works on the JAFFE dataset.

	JAFFE
Proposed method	98.44%
Reference [51]	92.8%
Reference [52]	73.28%
Reference [53]	94.83%
Reference [54]	96.30%
Reference [55]	97.46%

**Table 2 sensors-23-04770-t002:** Performance comparison between our proposed pipeline with previous works on the MMI database.

	MMI
Proposed method	99.02%
Reference [56]	98.63%
Reference [57]	82.74%
Reference [58]	83.41%
Reference [59]	83.56%

**Table 3 sensors-23-04770-t003:** Performance comparison between our proposed pipeline and previous works on the FER2013 dataset.

	FER2013	Extra Training Data
Proposed method	74.64%	✗
Reference [60]	75.97%	✓
Reference [61]	64.46%	✗
Reference [62]	69.57%	✗
Reference [63]	70.04%	✗
Reference [64]	74.59%	✗
Reference [6]	72.03%	✗
Reference [52]	73.28%	✗
Reference [65]	74.14%	✗
Reference [66]	72.16%	✗
Reference [67]	75.42%	✓
Reference [47]	76.82%	✓

**Table 4 sensors-23-04770-t004:** Performance comparison between our proposed pipeline and previous works on the AFFECTNET dataset.

	AFFECTNET 8 Emotions	AFFECTNET 7 Emotions	Extra Training Data
Proposed method	62.78%		✗
Reference [68]		66.46%	✗
Reference [69]		66.37%	✓
Reference [70]	65.20%		✗
Reference [6]	63.36%		✗
Reference [71]	63.03%	66.29%	✗
Reference [72]	63.00%		✗
Reference [73]	62.09%	65.69%	✗
Reference [74]	61.60%	65.40%	✓
Reference [75]	61.32%	65.74%	✓
Reference [76]	53.93%		✗
Reference [77]	59.30%		✗

**Table 5 sensors-23-04770-t005:** Performance comparison between different training methods on the FER2013 dataset.

	FER2013
Training with Linear Kernel SVM	74.64%
Training with with RBF kernel SVM	74.52%
Training with with Polynomial kernel SVM	74.50%
Training with with Sigmoid kernel	74.49%
Training with LDA	74.62%
Training with Softmax Regression	74.53%
Training with Cross-Entropy Loss	71.40%
Training without Weighted Data Sampling method	64.56%

**Table 6 sensors-23-04770-t006:** Performance comparison between different training methods on the AFFECTNET dataset.

	AFFECTNET
Training with Linear Kernel SVM	62.78%
Training with LDA	62.76%
Training with Softmax Regression	62.73%
Training with Cross-Entropy Loss	60.70%
Training without Weighted Data Sampling method	51.32%

## Data Availability

Not applicable.

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
