# Peer review of "Robust Human Face Emotion Classification Using Triplet-Loss-Based Deep CNN Features and SVM"

_sensors, 2023, doi:10.3390/s23104770_

Round 1

Reviewer 1 Report

The article discusses aspects related to deep machine learning technologies for solving problems associated with the automatic recognition of people's emotions based on the visual analysis of their facial characteristics. This topic is undoubtedly relevant. The authors propose their approach, which, according to them, can provide a good technical basis for recognizing people's emotions visually. Overall, the article is clear, and many points are described in detail. Additionally, there are many illustrations, graphs, and tables, which is also a plus. However, I would like to point out some shortcomings that, in my humble opinion, need to be corrected.

Shortcomings:

1) The first thing that strikes the eye is the fact that the authors of the paper in section 2 do not describe the current and, importantly, the best emotion recognition results that were obtained on the datasets used. For example, paperwithcode has a SOTA of the best accuracies for each corpus and it makes sense to add a description of, for example, the best (first) 3 results for each dataset (for the 7 and 8 basic emotions).

SOTA for AffectNet (7 emotions): https://paperswithcode.com/sota/facial-expression-recognition-on-affectnet (Emotion-GCN, EmoAffectNet, Multi-task EfficientNet-B2). SOTA for AffectNet (8 emotions): Multi-task EfficientNet-B2 (repeated), MT-ArcRes, DAN.

SOTA for FER2013: https://paperswithcode.com/sota/facial-expression-recognition-on-fer2013 (Ensemble ResMaskingNet with 6 other CNNs, Segmentation VGG-19, Local Learning Deep+BOW).

This will also extend references to previous work of the global scientific community (2020-23), which, are constantly presented at highly rated conferences focused on multimodal video analysis work (CVPR, ICCV, ECCV, FG, ICASSP, SPECOM and others) or corpus collection (LREC and others, as well as in Q1 level journals (Neurocomputing and others).

2) It would be helpful to explain why the Attention Module was not utilized in the paper.

3) Was modern MixUp techniques employed in the study? If yes, provide a detailed description. If not, why was it not used?

4) The paper did not use Cosine annealing during training. What was the reason for this omission?

5) Elaborate more on the keywords used in the article.

6) Include a paragraph in the conclusion discussing potential future work and research areas.

7) The style of the article needs to be revised to address the spelling and punctuation errors present. Despite this, the article remains easy to read.

As it stands, the article is shortcomings and should be improved. However, it should be replaced that the article is easy to read.

Author Response

Dear Reviewer,

Thank you for taking the time to review my manuscript. Please find attached the responses to your comments. I have addressed all of your concerns and made revisions accordingly. If you have any further questions or suggestions, please do not hesitate to contact me.

Thank you again for your valuable feedback.

Best regards,

Reviewer 2 Report

The manuscript entitled “Robust human face emotion classification using triplet loss based deep CNN features and SVM” has been investigated in detail. The topic addressed in the manuscript is potentially interesting and the manuscript contains some practical meanings, however, there are some issues which should be addressed by the authors:

1)      In the first place, I would encourage the authors to extend the abstract more with the key results. As it is, the abstract is a little thin and does not quite convey the interesting results that follow in the main paper. The "Abstract" section can be made much more impressive by highlighting your contributions. The contribution of the study should be explained simply and clearly.

2)      The readability and presentation of the study should be further improved. The paper suffers from language problems.

3)      The “Introduction” section needs a major revision in terms of providing more accurate and informative literature review and the pros and cons of the available approaches and how the proposed method is different comparatively. Also, the motivation and contribution should be stated more clearly.

4)      The importance of the design carried out in this manuscript can be explained better than other important studies published in this field. I recommend the authors to review other recently developed works.

5)      “Discussion” section should be added in a more highlighting, argumentative way. The author should analysis the reason why the tested results is achieved.

6)      The authors should clearly emphasize the contribution of the study. Please note that the up-to-date of references will contribute to the up-to-date of your manuscript. The study named- “Artificial intelligence-based robust hybrid algorithm design and implementation for real-time detection of plant diseases in agricultural environments; Detection of solder paste defects with an optimization‐based deep learning model using image processing techniques; Optimization of deep learning model parameters in classification of solder paste defects” - can be used to explain the optimization method in the study or to indicate the contribution in the “Introduction” section.

7)      How to set the parameters of proposed method for better performance?

8)      The complexity of the proposed model and the model parameter uncertainty are not enough mentioned.

9)      It will be helpful to the readers if some discussions about insight of the main results are added as Remarks.

10) Please see some additional question in the file attached.

This study may be proposed for publication if it is addressed in the specified problems.

Author Response

(The authors gave the same response as above.)

Round 2

Reviewer 1 Report

The authors of the article practically finalised the article. However, there is one other disadvantage which in my opinion will mislead readers.

In my previous review there was a comment under number 1. The authors of the article did add the necessary tables. However, if you look at the added table 4, in its current form the best method for recognizing 7 basic emotions on the AffectNet corpus is Multi-task EfficientNet-B2 method (IEEE Transactions on Affective Computing journal). In fact, this method is not currently the best, but only the third best method in SOTA on paperswithcode: https://paperswithcode.com/sota/facial-expression-recognition-on-affectnet?metric=Accuracy%20(7%20emotion). The best ones are Emotion-GCN  (66.46%, 1st place, presented at the highly rated Automatic Face and Gesture Recognition conference, FG 2021), EmoAffectNet (66.37%, 2nd place, presented in Q1, Neurocomputing 2022).

I can of course assume that the authors did not indicate the best work on the 7 basic emotions because these methods have not been tested on the 8 emotions. However, in any case, this does not exactly turn out to be reliable information in Table 4. That's why we need to add the best methods.

I think that the authors of this article will agree with me that it is necessary to provide always up-to-date information about the best methods, especially those presented in important conferences and journals.

I suggest correcting this shortcoming and then the article could be accepted for publication and useful for specialists who link their research to machine learning and emotion recognition.

Author Response

Dear Reviewer,

Thank you for your thorough review of our article. We appreciate the time and effort you have put into providing valuable feedback that will help improve the quality of our work.

In response to your comments, we have carefully revised the manuscript and made changes according to your suggestions. We have also prepared a detailed response file that addresses of your comments and explains how we have addressed them in the revised version.

Please find attached the revised manuscript along with the response file. We hope that these changes and explanations will satisfactorily address your concerns and that you find the revised version to be an improvement over the original submission.

Once again, thank you for your valuable feedback, and we look forward to hearing your thoughts on the revised version.

Reviewer 2 Report

All my comments have been thoroughly addressed. It is acceptable in the present form.

Author Response

Dear Reviewer,

Thank you for taking the time to review our article and for your positive feedback regarding the revisions we made. We are pleased to hear that all of your comments have been thoroughly addressed and that the revised version is acceptable in its present form.

Your thoughtful and constructive comments have greatly improved the quality of our article, and we are grateful for your insights and suggestions. We appreciate your time and effort in reviewing our work, and we look forward to the possibility of working with you again in the future.

Thank you once again for your valuable feedback and for your contribution to the advancement of our field.

Round 3

Reviewer 1 Report

This article looks complete. All comments and shortcomings have been successfully fixed by the article authors. All questions have been answered. The article is ready for publication.